# Role of NF-κB in Platelet Function

**DOI:** 10.3390/ijms20174185

**Published:** 2019-08-27

**Authors:** Kevin Kojok, Abed El-Hakim El-Kadiry, Yahye Merhi

**Affiliations:** 1The Laboratory of Thrombosis and Hemostasis, Montreal Heart Institute, Research Centre, 5000 Belanger Street, Montreal, H1T 1C8, QC, Canada; 2Faculty of Medicine, Université de Montréal, Montreal, H3T 1J4, QC, Canada

**Keywords:** NF-κB, platelet, signaling pathways, inflammation, thrombosis

## Abstract

Platelets are megakaryocyte-derived fragments lacking nuclei and prepped to maintain primary hemostasis by initiating blood clots on injured vascular endothelia. Pathologically, platelets undergo the same physiological processes of activation, secretion, and aggregation yet with such pronouncedness that they orchestrate and make headway the progression of atherothrombotic diseases not only through clot formation but also via forcing a pro-inflammatory state. Indeed, nuclear factor-κB (NF-κB) is largely implicated in atherosclerosis and its pathological complication in atherothrombotic diseases due to its transcriptional role in maintaining pro-survival and pro-inflammatory states in vascular and blood cells. On the other hand, we know little on the functions of platelet NF-κB, which seems to function in other non-genomic ways to modulate atherothrombosis. Therein, this review will resemble a rich portfolio for NF-κB in platelets, specifically showing its implications at the levels of platelet survival and function. We will also share the knowledge thus far on the effects of active ingredients on NF-κB in general, as an extrapolative method to highlight the potential therapeutic targeting of NF-κB in coronary diseases. Finally, we will unzip a new horizon on a possible extra-platelet role of platelet NF-κB, which will better expand our knowledge on the etiology and pathophysiology of atherothrombosis.

## 1. Introduction

Cardiovascular disease is the largest cause of death globally. According to the report from the World Health Organization (WHO), the death rate caused by cardiovascular diseases worldwide is estimated at 17.3 million people in 2008, accounting for 30% of deaths globally. In addition, it is estimated that this number may increase to 23.6 million people [1]. Cardiovascular disease is often caused by thrombotic events such as coronary heart disease. Platelets are major players in the occurrence of cardiovascular diseases since they are involved in various thrombo-inflammatory diseases, particularly atherosclerosis and its progression to atherothrombosis in acute coronary syndrome (ACS) patients [2,3,4,5]. Indeed, platelets mediate primary hemostasis and formation of thrombosis, however, thrombosis can become pathological when it occurs, mainly, after the rupture of an atherosclerotic plaque. Thus, following an atherosclerotic vascular lesion, exposed sub-endothelial collagen, alongside thrombin and adenine diphosphate (ADP) production, triggers platelet activation by interacting with several receptors expressed on platelets such GPIb-IX-V, GPVI-FcRγ, protease activated receptors (PARs), P2Ys, and integrins (α_2_β_1_, α_5_β_1_, α_6_β_1_, α_v_β_3_ et α_IIb_β_3_). Consequently, the “inside-out” signaling of the α_IIb_β_3_ integrin, in response to platelet activation, leads to platelet aggregation thanks to the binding of α_IIb_β_3_ to fibrinogen promoting the formation of a thrombus [6,7]. Such a process could induce a partial or complete occlusion of the blood vessel, which leads to a decrease or blockage of the blood flow and thus becomes a cause of occurrence of ischemia or infarction of an irrigated organ such as the heart. Furthermore, the release of a plethora of inflammatory mediators by activated platelets such as soluble P-selectin, soluble CD40 ligand (sCD40L, also known as CD154) and interleukin-1 beta (IL-1β), which interact with cells that mediate inflammation such as circulating leukocytes, endothelial cells, and progenitor cells, and thereby aggravate inflammatory responses [8,9]. At the molecular level, several pathways promoting the initiation and progression of inflammatory diseases [10] are regulated by NF-κB, a prominent transcription factor extensively described in innate and adaptive immune cells as a governor of essential physiological processes including cell survival, proliferation, and activation. In terms of atherosclerosis, vascular fatty plaques host a pro-inflammatory milieu partly kindled by platelets in which NF-κB appears as an important regulator of inflammatory and thrombotic responses, albeit to a less clear extent in terms of mechanics and functions, compared to NF-κB expressed in various cells [11]. Indeed, the activation of NF-κB in endothelial cells in response to an inflammatory environment triggers the expression of adhesion molecules, which increases binding and transmigration of leukocytes and platelets, while unleashing their thrombogenic potential. The activation of NF-κB in monocytes is required for their differentiation to macrophages and contributes to the expression of tissue factor and release of inflammatory cytokines. In neutrophils, NF-κB activation extends their survival and induce the expulsion of neutrophil extracellular traps (NETs), which exert antibacterial functions and triggers a strong coagulatory response and may induce the formation of microthrombi. NF-κB plays a crucial role in lymphocyte proliferation and cytokines production, which promotes enhanced inflammatory and thrombotic response by recruiting platelets [11]. In this review, since NF-κB is a central hub in thrombo-inflammatory reactions, we aim to enhance our knowledge on the less-characterized role of NF-κB in platelets. Moreover, we will harness this novel non-genomic role of NF-κB at in pinpointing its implications in cardiovascular diseases like atherosclerosis and atherothrombosis in ACS. Furthermore, we will highlight the effects of several active compounds on NF-κB, thereby bridging to its potential therapeutic utility particularly at the level of platelets. Finally, we will widen our compass to include a potential extra-platelet role of platelet NF-κB, which might oblige us to modify our understanding not only of the role of platelets but also of the progression of atherothrombotic diseases.

## 2. The Genomic Role of NF-κB

NF-κB proteins exist in the cytoplasm as inactive dimers associated with the inhibitor IκB subunits (IκBα and IκBβ) which prevent their activation. Of the two subunits, IκBα is the most represented. In most cells, those dimers are formed by five Rel/NF-κB DNA-binding subunits: p50 NFκB1, p65/REL A, cRel, NFκB2/p52, and Rel B. NF-κB is regulated by the multi-subunit IκB kinase (IKK) [12], which catalyzes the phosphorylation of IκB causing its proteasome-mediated degradation. Among IKK subunits, IKKβ is the most active. Indeed, IKKβ deficiency in embryonic mice fiercely prevents NF-κB activation [13]. Once IκB is degraded, active NF-κB is liberated, after which it translocates to the nucleus to transcribe pro-inflammatory and pro-survival genes in two distinctive pathways, canonical and non-canonical, implicating p50/RelA and p52/RelB, respectively [14] (Figure 1). In such cellular settings, the role of NF-κB is genomic and well-documented since its discovery in immune cells over 30 years ago [15,16,17,18,19,20,21,22,23].

## 3. NF-κB Expression in Platelets

Acellular fragments are derived from megakaryocytes, platelets are devoid of nuclei, yet they express IKK, IκB, and NF-κB [24]. Likewise, NF-κB is expressed in other anucleated cells like mature erythrocytes [25]. Specifically, Liu et al. were the first to demonstrate the expression of NF-κB in platelets in 2002, revealing that thrombin-induced platelet activation triggers the degradation of IκBα following its serine 32 residue phosphorylation [26]. Afterwards, several other studies corroborated the same finding [24,27,28,29], suggesting that platelets have a much more complex nature than simply being the “remnants of megakaryocytes”. This is supported by the identification of several other transcription factors in platelets such as peroxisome proliferator–activated receptors (PPARs) [30,31,32,33], retinoid X receptor (RXR) [34,35], glucocorticoid receptor (GR) [36,37], STAT3 [38], as well as spliceosomes, transcriptomes, messenger RNAs (mRNAs), microRNAs (miRNAs) [39,40,41,42], and other diverse components, which partake in translational execution in a signal-driven manner [43]. The exact role of platelet NF-κB, however, remains elusive.

## 4. NF-κB Functions in Platelets

As is the case in other cells, NF-κB signaling in platelets involves IKKβ phosphorylation, IκBα degradation, and p65 phosphorylation [24,44,45]. However, unlike other cells, the culminating events of NF-κB signaling in platelets remain partially understood. For this purpose, multiple studies were performed utilizing either pharmacological inhibitors of NF-κB such as BAY 11-7082 or knockout mice to unravel the role of NF-κB in platelets, specifically in the context of their survival/apoptosis, priming, activation, and aggregation.

### 4.1. Platelet Survival and Apoptosis

The least ventured in platelet NF-κB research is its function in survival and apoptosis, namely because platelets are short-lived with a lifespan close to 5 days in mice and 10 days in humans [46]. A report by Dowling et al. reasons out this brief lifespan using a bio-mathematical model, demonstrating that early platelet senescence is an internally controlled mechanism rather than being the result of multiple deteriorating hits [47]. More specifically, it is proposed that early after shedding from megakaryocytes, platelets encompass enough B-cell lymphoma-extra large (Bcl-xL), a protein of the antiapoptotic Bcl-2 family, to overthrow the effects of Bax and Bak, pro-apoptotic molecules, which disrupt mitochondrial membrane permeability and trigger caspase-driven apoptosis. Later in their lifespan, however, and in the absence of external death signals, platelets are unable to synthesize more Bcl-xL, and thus the more labile Bak and Bax regain their status, driving platelets to their final demise-being washed away from the blood stream [48]. Although the proteasome might be a major contributor [49], the players and exact mechanisms that control this putative internal timer between life and death are still partially understood, especially given that caspase-independent pathways are also recorded in platelets [49,50]. In multiple cell types, NF-κB confers a pro-survival role by promoting G1-to-S phase cell cycle progression [51], inducing the transcription of several above-mentioned cytokines and growth factors, and upregulating anti-apoptotic proteins (XIAP, Bcl-xL, Bcl-2, A1/Bfl-1) [52,53]. In platelets, however, the role of NF-κB in the same context is still elusive. A recent study showed that upon treatment of platelets with NF-κB inhibitors, a significant increase of intracellular calcium was recorded in parallel with (i) decreased sarco/endoplasmic reticulum (ER) Ca^2+^-ATPase (SERCA) activity, (ii) increased ER stress, (iii) pronounced mitochondrial membrane depolarization and mitochondrial permeability transition pore (MPTP) formation, (iv) downregulated Bcl-2 levels, and (v) increased caspase activity and apoptosis. Upon pharmacologically preventing ER stress, however, MPTP formation and apoptosis were reversed. This string of events suggests that NF-κB might moderate calcium hemostasis in platelets, possibly by regulating the function of ER membrane-implemented SERCA, thereby favoring survival and preventing ER stress-triggered mitochondrial-driven platelet apoptosis [54]. Aside from the apoptosis pathway, inhibiting GPIbα shedding in platelets was able to reduce platelet clearance [55,56,57]. Indeed, the shedding of GPIbα is a physiological mechanism mediated by a disintegrin and metalloproteinase domain-containing protein 17 (ADAM17) that takes place constantly on the platelet surface [58]. However, GPIbα shedding is linked to an increase in platelet clearance. Indeed, under physiological shear stress, the binding of Von Willebrand Factor (VWF) with GPIbα induces unfolding of the mechanosensory domain (MSD) on the platelet surface, thereby boosting shedding of GPIbα and triggering GPIb-IX signaling, leading to rapid platelet clearance [59,60]. NF-κB could contribute to GPIbα shedding in platelets; for instance, in response to thrombin stimulation, IKKβ-deficient platelets were unable to shed GPIbα. However, in response to ADP or collagen stimulation, GPIbα shedding was unaffected; this might suggest that IKKβ is uniquely implicated in thrombin-induced GPIbα shedding [61].

### 4.2. Platelet Activation and Priming

Following their stimulation with specific agonists such as thrombin or collagen, platelets undergo activation in a series of processes including shape change, cytoskeleton rearrangement, and organelle centralization. In parallel, the release of dense granules content including ADP occurs, triggering further platelet recruitment and activation. More specifically, ADP activates platelets by binding its receptors, P2Y_1_ and P2Y_12_, and inducing the expression of platelet P-selectin. Similarly, the release of thromboxane A2 (TXA_2_) enhances platelet activation and recruitment and leads to platelet aggregation [62,63].

NF-κB inhibitors prevent platelet activation following stimulation with several agonists, suggesting that NF-κB in platelets functions rather non-genomically inducing platelet activity.

#### 4.2.1. Thrombin-Activated Platelets

Thrombin is a key activator of human platelets by binding PAR-1 and PAR-4 receptors, promoting the expression of several inflammatory mediators such as P-selectin, IL-1β, and CD40L [64]. Moreover, thrombin triggers ADP release, TXA_2_ production, and IκBα phosphorylation [26]. Thrombin-stimulated platelets pretreated with BAY 11-7082, an irreversible inhibitor of IKKβ phosphorylation, or Ro 106-9920, a selective inhibitor of IκBα ubiquitination, exhibit drastically low expression of P-selectin, TXA_2_ production, and ADP release [27]. Interestingly, the phosphorylation of ERK, a mediator of granule secretion, is inhibited, too [65]. Karim et al. further explain IKKβ implication in platelet secretion through its ability to phosphorylate synaptosome-associated protein-23 (SNAP-23), a member of membrane proteins complex called soluble *N*-ethylmaleimide-sensitive-factor attachment protein receptors (SNAREs), which regulates granule secretion. Indeed, SNAREs play a major role in the fusion of platelet granules with the platelet membrane. In fact, when platelets are activated in response to thrombin stimulation, vSNAREs (vesicular SNAREs) such as VAMP-8 (vesicle-associated membrane protein–8) that present on platelet granules surface bind to tSNAREs (target SNAREs) such as Syntaxin 11 expressed on the platelet membrane. The binding is mediated mainly by SNAP-23. Thus, SNAP-23 phosphorylation in thrombin-stimulated platelets facilitates the fusion between granules and plasma membrane for cargo release [66]. In contrast, the use of selective inhibitors of IKKβ (BMS-345541 [67], TPCA-1 [68], and BAY-11-7082) prevents thrombin-stimulated SNAP-23 phosphorylation in a dose-dependent manner [66]. Additionally, the same group shows that thrombin-stimulated platelets from IKKβ knockouts have a lower capacity to release alpha, dense, and lysosomal granules content by an overall decrease of 30%. Wei et al. [61], who demonstrated that platelet IKKβ deficiency exhibits decreased secretion and activation confirmed this observation. Interestingly, a recent study showed that the inhibition of MALT1, an upstream regulator of IKK complex and a member of the functional proteasome CARMA/MALT1/Bcl10 (CBM) complex, prevented platelet activation and secretion by abrogating SNARE formation [69]. Such findings suggest that NF-κB could contribute to platelet degranulation in platelet secretion.

It is suggested that PAR-4, a thrombin receptor, is the main activator of NF-κB in platelets through a mechanism involving sphingomyelinase (nSMase), an enzyme which catalyzes the transformation of sphingomyelin into ceramide and phosphorylcholine and which notably partakes in macrophage NF-κB activation [70]. More specifically, the binding of thrombin to platelet PAR4 induces nSMase activation, which increases C24:0-ceramide levels. This is followed by the activation of p38 MAPK, which in turn initiates NF-κB and platelet activation [71].

#### 4.2.2. Collagen-Activated Platelets

In addition to thrombin, collagen is another platelet agonist discovered to phosphorylate IκBα and thus activate NF-κB [24]. Collagen interacts with platelet glycoprotein receptors, GP1b and GPVI, triggering intracellular signaling, promoting integrin αIIbβ3 receptor activation, and inducing the release of secondary mediators like ADP and TXA_2_. The use of BAY-11-7082 prior to platelet stimulation with collagen elevates cyclic AMP and enhances vasodilator-stimulated phosphoprotein (VASP) phosphorylation, therefore, suppressing TXA_2_ formation, ATP release, P-selectin expression, and intracellular Ca^2+^ immobilization [29]. Of note, VASP, an actin-and prolifin-binding protein and a substrate of cAMP-dependent protein kinase A (PKA) [72] is a major negative regulator of platelet secretion and adhesion [73]. Oppositely, a study by Gambaryan et al. suggested that upon IκBα degradation, PKA dissociates from NF-κB and phosphorylates VASP. In addition, following platelet pre-treatment with an IKK inhibitor VII, a competitive reversible inhibitor of IKKβ, VASP phosphorylation, and its platelet-inhibitory effect is lost after stimulation with both, collagen and thrombin [28]. However, this study might be demonstrating a peculiar negative feedback signaling to avert excessive platelet activation. Nevertheless, the results require further explanation.

#### 4.2.3. CD40L-Primed Platelets

As mentioned before, platelet agonists trigger the expression of several inflammatory mediators, among which is CD40L, a member of the TNF family mainly expressed in T lymphocytes and activated platelets. CD40L rapidly appears on the platelet surface following activation, upon which it is subsequently cleaved generating a soluble fragment of 18-kDa, termed sCD40L [74]. Activated platelets constitute the primary source of sCD40L, accounting for >95% of its plasmatic concentration [75,76]. Elevated levels of sCD40L are now considered reliable predictors of cardiovascular diseases [76,77,78,79,80,81,82,83,84,85]. The discovery of new CD40L receptors (α_IIb_β_3_, α_5_β_1_, and α_M_β_2_) [86,87,88], in addition to its classical/high-affinity receptor CD40 [89], adds complexity to the diverse interplays to which CD40L takes part in cells in general and platelets in particular.

The receptor CD40, constitutively expressed on platelets, lacks intrinsic signaling activity and needs to recruit adaptor molecules, such as the (TNFR)-associated factors (TRAFs) that bind to the cytoplasmic domain of CD40 and subsequently recruit kinases and other effector proteins responsible for transducing signals [90]. CD40 can bind five of the seven TRAF family members (TRAF1, 2, 3, 5, and 6). The cytoplasmic domain of CD40 has a proximal TRAF6 binding site and a more distal TRAF2/3/5 binding site. TRAF1 only binds to CD40 when CD40 signaling is already active and acts as a regulator rather than an activator of CD40 signaling [91]. The binding of TRAF5 to CD40 is still debated, and its role in CD40-mediated signaling remains controversial [92]. Overall, CD40-TRAF signals stimulate kinase activation and gene expression and induce the production of antibodies and a variety of cytokines, expression and upregulation of adhesion molecules, and protection or promotion of apoptosis. These various pathways can culminate in either the induction or inhibition of biological functions.

Conversely, platelets express the receptors α_IIb_β_3_ and α_5_β_1_ in their inactive form, whereupon platelet activation by ligands such as fibrinogen and fibronectin, undergo certain conformational changes including the interaction through their cytoplasmic tails with intracellular signal transduction proteins such as as FAK, Src, and talin rendering them active [93]. CD40L can bind both inactive and active forms of α_IIb_β_3_, yet can only bind the inactive form of α_5_β_1_ [94]_._

In platelet physiology, our laboratory showed that CD40L alone induced IκBα phosphorylation and NF-κB activation exclusively through platelet CD40 receptor [95]. We also showed that sCD40L, in the presence of sub-optimal doses of platelet agonists like collagen and thrombin, significantly increased platelet activation and aggregation through an NF-κB-independent CD40/TRAF-2/Rac-1/p38 MAPK axis [9]. The first result explains the priming effects of CD40L, priming being a pre-activation process that ultimately prepares platelets for aggregation through a series of molecular events including P-selectin expression and lamellepodia formation, all of which we found concurrent with NF-κB activation. The second result, however, might explain that NF-κB activity driven by sCD40L priming capacity bridges to CD40/TRAF-2/Rac-1/p38 MAPK axis-induced platelet aggregation. This was evident in first, following the use of IKK inhibitor VII, platelet priming, and potentiation of aggregation were significantly diminished, and second, IκBα and p38 MAPK phosphorylation were found to be independent upon platelet pre-treatment with either of their inhibitors followed by sCD40L stimulation. Furthermore, our recent study dug deeper into the unique NF-κB signaling pathway, showcasing that CD40L activated NF-κB through CD40 and TAK-1 and endorsing that TAK-1 is upstream of NF-κB in CD40L signaling as its inhibition completely eliminated NF-κB activation [95]. Nevertheless, a study by Kuijpers et al. [96] contradicted our results, demonstrating that CD40L enhanced collagen-induced platelet-platelet interactions by supporting integrin αIIbβ3 activation, platelet secretion, and thrombus growth via PI3Kβ but not CD40 and IKKα/NFκB. Still and all, the paper claimed that first, CD40 deficiency led to increased integrin αIIbβ3 expression, which contradicts the results of a third study by Inwald et al. [97] and second, P-selectin expression and phosphatidylserine exposure were not affected by CD40L preincubation, thereby also contradicting several other studies [9,97,98,99].

In summary, Figure 2 resumes our previous findings that established the link between sCD40L, enhanced platelet reactivity and thrombosis involving its main receptor CD40 binding to an adaptor protein the (TNFR)-associated factor-2 (TRAF2) and downstream signaling via Rac1/p38-MAPK [9]. We have also revealed that sCD40L is a potent activator of NF-κB [100], which primes platelets through CD40 signaling via the transforming growth factor-B (TGF-B)-activated Kinase (TAK1) [95].

#### 4.2.4. TLR Ligand-Activated Platelets

Involved in immune and inflammatory responses, the family of toll-like receptors (TLRs) is expressed on platelets [101,102]. Each TLR identifies different types of pathogen-associated molecular patterns (PAMPs) or ligands present on viruses, bacteria, and fungi. Among TLRs, TLR2, and TLR4 are the most involved in platelet activation, as translated by their increased expression [103]. Following platelet TLR 1/2 stimulation with Pam3CSK4, a synthetic agonist of TLR2/TLR1, αIIbβ3 activation, and P-selectin expression are increased, leading to hemostatic and inflammatory responses. This coincides with an elevation in PI3K/Akt, ERK1/2, and p38 activity, P2 × 1-mediated Ca^2+^ mobilization, TXA_2_ production, and ADP release. As for TL4, platelet activation with lipopolysaccharide (LPS), a TLR4 agonist, triggers platelet secretion and potentiates platelet aggregation via TLR4/MyD88 and cGMP-dependent protein kinase pathways [104]. It is also well documented that TLRs activate NF-κB in nucleated cells and drive the production of proinflammatory cytokines like IL-1β and TNFα [105,106]. A study by Rivadeneyra et al. demonstrated that TLR2 and 4 agonists induced platelet activation responses through NF-κB. In parallel, IκBα degradation and p65 phosphorylation were observed. In a relatable manner, platelet treatment with BAY 11-7082 or Ro 106-9920 impaired TLR-mediated platelet activation [44]. Other studies on chicken thrombocytes, the hematological equivalents to mammalian platelets, highlighted the role of NF-κB in TLR signaling, showing that inhibition of IKK with BMS345541 results in a significant reduction in thrombocytes’ secretory profile [107,108].

#### 4.2.5. AGE-Activated Platelets

The binding of advanced glycation end-products (AGEs) to their receptor, RAGE, is believed to play an important role in the pathophysiology of several cardiovascular diseases such as heart failure and coronary disease, as well as peripheral artery diseases observed in diabetes [109]. RAGE is expressed on a plethora of cells such as macrophages, monocytes, endothelial cells, neutrophils, lymphocytes, and platelets [110,111,112,113]. In platelets, RAGE activation increases P-selectin expression on the platelet membrane, hence promoting platelet activation [111]. Knowing that RAGE induces NF-κB activation leading to the secretion of pro-inflammatory cytokines in monocytes [114] and that platelets possess all elements of NF-κB signaling cascade downstream RAGE, including ERK and p38 MAP kinase [115], it is possible that RAGE might induce NF-κB activation in platelets, however, further investigations are required to confirm this theory.

#### 4.2.6. Epinephrine-Primed Platelets

Secreted by the adrenal gland, epinephrine is a hormone with alpha- and beta-adrenergic sympathomimetic activities [116,117]. Alone, epinephrine does not induce platelet activation through platelet alpha 2-adrenergic receptors. However, when present with other platelet agonists, it can potentiate their activation and aggregation responses, hence being a platelet “primer” like CD40L mentioned above [118]. This priming action is primarily mediated by IKK activation, which culminates in inducing p38 and PKA signaling as well as activating α_IIb_β_3_ integrin receptor [11]. In platelet rich plasma pre-treated with BAY 11-7082 and Ro 106-9920, the potentiating action of epinephrine is impaired in the initial stages of aggregation inclusively, further evidencing the involvement of NF-κB in moderating the early stages of platelet activation [27].

#### 4.2.7. ADP-Activated Platelets

ADP is the earliest identified platelet agonist which exerts its effects through three purinergic receptors, two of which are G protein-coupled: (i) G_αq_ coupled-P2Y1, (ii) G_αi_-coupled P2Y12, and Ca^2+^ channel P2 × 1 [119]. Some of the actions mediated by ADP on platelets include adhesion, shape change, granule secretion, Ca^2+^ influx and intracellular mobilization, adenylyl cyclase inhibition, TXA_2_ production, and aggregation induction [120,121]. The role of NF-κB in regulating the initial stages of ADP-induced platelet activation is demonstrated upon pre-incubation of platelet rich plasma with BAY 11-7082 and Ro 106-9920 followed by ADP treatment. Resulting is a significant impairment of early platelet aggregation [27].

### 4.3. Platelet Aggregation

Platelet aggregation is the ultimate process in primary hemostasis, wherein platelets clump together creating a stable hemostatic plug following their endothelial damage. The cross-linking of essentially activated αIIbβ3 integrin receptors on adjacent platelets by soluble fibrinogen mediates this process. In fact, platelets express an inactive form of αIIbβ3 integrin, which upon platelet activation by thrombin, collagen, and ADP, intracellular signals are triggered, leading to conformational changes in αIIbβ3 that transit the receptor from an inactive low affinity state to an active high affinity state. This “inside-out” signaling mechanism allows the full exposure of fibrinogen binding sites in αIIbβ3 integrin.

The use of inhibitors of IKKβ phosphorylation highlighted the influence of NF-κB activation on platelet aggregation. Therein, treating platelets with BAY 11-7082 or Ro 106-9920 decreases fibrinogen binding following thrombin and collagen stimulation, which inhibits αIIbβ3 inside-out signaling, platelet aggregation, and clot retraction [27,29,71]. Andrographolide, an active ingredient found in the leaves of a medicinal herb, is another less-utilized potent NF-κB inhibitor, which prevents p65 phosphorylation in collagen-stimulated platelets NF-κB and interferes with their function. Although andrographolide is shown to inhibit platelet aggregation via eNOS activation and inhibition of both, PLCγ2–PKC and PI3kinase/Akt-MAPKs pathways, it is unknown yet if this is the result of its NF-κB inhibitory functions [45,122]. As aforementioned, NF-κB signaling has a significant role in platelet activation. Therefore, it is not surprising that the cytoskeletal rearrangements leading to the conformational changes of αIIbβ3 integrin, which grants platelets their activity, do not occur since IKKβ phosphorylation inhibitors dampen platelet activation. Our laboratory showed that the use of IKK inhibitor VII and BAY 11-7082 prevented aggregation of platelets pretreated with sCD40L and stimulated thereafter with a priming dose of collagen [100]. Surprisingly, the study by Gambaryan et al. demonstrated the opposite role of IKKβ inhibitors, which they potentiated platelet aggregation. Nevertheless, the maximum amplitude of aggregation was similar between IKK inhibitor VII-treated platelets and controls following their stimulation with optimal doses of thrombin and collagen. Therefore, a more plausible result would necessitate the stimulation of IKK inhibitor VII-pretreated platelets with a priming/sub-optimal dose of those agonists, in order to better observe the potentiating effect of the inhibitor. In this context, our recent study demonstrated that the aggregation of platelets pretreated with BAY 11-7082 or TAK-1 inhibitors, 5Z-7-Oxozeaenol and Takinib, primed with sCD40L, and then stimulated with sub-optimal doses of thrombin was abrogated completely [95]. Our results also pinpointed the emerging importance of TAK-1 in NF-κB activation in platelets as another attractive therapeutic target in thrombotic diseases.

Similarly, an in vivo study by Karim et al. reported that the inhibition of IKKβ in mice slowed down thrombus formation and increased bleeding time. Moreover, IKKβ knockout mice also showed increased bleeding times, consistent with the effects of pharmacological inhibition [66]. Of note, Wei et al. demonstrated that IKKβ deficiency promoted leukocyte-platelet interaction by delaying ADAM17-mediated GPIbα shedding, therefore, enhancing platelet aggregation [61].

The pro-coagulant activity of NF-κB was also evident in thrombo-inflammatory conditions like sepsis, in which tissue factor was upregulated in parallel with a downregulation of anticoagulation molecules like anti-thrombin and tissue factor pathway inhibitor. Eventually, the pronounced production of thrombi in this condition induces platelet depletion and subsequent bleeding vulnerabilities, which reflects an extreme role of NF-κB in hematological diseases [123,124].

In the context of diabetes, a recent study on diabetic rats and humans demonstrated that the enhanced expression of platelet P2Y_12_, a pro-thrombotic ADP receptor, correlates with increased IκBα phosphorylation and degradation in platelets and megakaryocytes as well as increased p65 expression and binding to P2Y_12_ promoter in megakaryocytes. The study hence suggested that hyperglycemic conditions render NF-κB the mediator of the increased expression of P2Y_12_ in platelets of patients with type 2 diabetes mellitus [125], evidencing further the pro-thrombotic role of platelet NF-κB in platelets.

Alongside NF-κB, other transcription factors regulate platelet aggregation. For instance, PPARγ agonists decrease CD40L and TXA_2_ release as well as platelet aggregation by modulating early GPVI receptor signaling and thus inhibiting collagen-mediated activation [126]. Moreover, RXR ligands inhibit TXA_2_ and ADP production and platelet aggregation by binding the GTP-binding protein Gq thus preventing its subsequent activation of Rac and aggregation-driving signaling pathways [34].

## 5. Natural/Pharmacological Compounds and NF-κB

Several studies on pharmacological and natural active compounds, even those specifically targeting platelets, recount off-target effects on cellular rather than platelet NF-κB.

Firstly, aspirin or acetylsalicylic acid (ASA) is a non-steroidal anti-inflammatory and anti-platelet drug. Requiring the lowest dose and used in the prevention of thrombosis, the anti-platelet indication is established through inhibiting the action of cyclooxygenase-1 (COX-1) responsible for TXA_2_ secretion [127]. Because of its dual function, ASA is the most studied medicament in terms of its effects on NF-κB. Kopp and Ghosh were the first to demonstrate that sodium salicylate, an ASA derivative, targets the NF-κB pathway by inhibiting IκB degradation and subsequent NF-κB nuclear translocation in activated T cell lines [128]. Later, it was shown in vitro as well that ASA and salicylate mediated the latter effects through IKKβ inhibition [129]. In vivo, in a rat model of acute pulmonary embolism, ASA reduced the expression of NF-κB in lung tissues dose-dependently [130]. In peripheral blood mononucleated cells (PBMC) of diabetic patients, the daily dosing of ASA significantly decreased NF-κB binding activity to DNA [131]. However, in atherosclerotic patients, NF-κB expression in carotid artery plaques, precisely that of foam and endothelial cells origins, was unaffected by ASA treatment [132], suggesting that ASA effects on NF-κB are cell- and/or pathophysiology-dependent. 

Platelet P2Y_12_ receptor antagonists such as ticagrelor and clopidogrel are another family of anti-platelet agents indicated for patients with acute coronary syndrome at risk of ischemic events [133]. In rats with gastric ulcer, ticagrelor impairs NF-κB p65 phosphorylation [134]. In porcine models undergoing coronary interventions, the long-term administration of clopidogrel shows significant reductions in NF-κB activity [135].

Another anti-aggregation agent, cilostazol is a phosphodiesterase (PDE) 3 inhibitor indicated for treating intermittent claudication in the lower periphery. By increasing cyclic adenosine monophosphate (cAMP) levels, it exhibits vasodilator as well as anti-platelet effects [136]. Cilostazol interferes with the transcriptional activity of NF-κB in macrophages after their treatment with TLR ligands, therefore, reducing the generation of TLR-mediated pro-inflammatory cytokines [137]. In platelets, a recent study on hypercholesterolemia rats treated with cilostazol reported enhanced IκBα expression coinciding with NF-κB inhibition, an effect linked to AMP kinase (AMPK) activation. The study went further by attributing to AMPK activation and subsequent NF-κB inhibition a major role in preserving endothelial function by reducing platelet P-selectin and CD40L expression while increasing endothelial nitric oxide synthase activity responsible for anti-platelet nitric oxide production [138]. Another non-selective PDE inhibitor with anti-inflammatory as well as anti-platelet functions, dipyridamole, exhibits inhibitory actions against IKKβ, IκB phosphorylation and degradation, and p65 nuclear translocation in macrophage cell lines [139] and human PBM [140].

Vorapaxar, a PAR-1 antagonist, recently indicated for the prevention of atherothrombotic events in patients with myocardial infarction and peripheral arterial disease [141], also shows off-label NF-κB effects. In endothelial cells stimulated with cholesterol, vorapaxar significantly increased NF-κB expression levels, suggesting that the molecule performs a protective role in atherosclerotic settings [142]. However, and although of a different clinical context pertaining to hypoxic effects on ventricular remodeling, another study showed the exact opposite, as endothelial cells from hypoxic mice treated with a PAR-1 antagonist demonstrates downregulated NF-κB levels. The same study showed similar results with rivaroxaban, an anticoagulant drug which functions by directly targeting the clotting factor Xa [143]. In another study, rivaroxaban was demonstrated to attenuate deep venous thrombosis in a rat model by targeting NF-κB signaling pathway in endothelial cells, more specifically downregulating IκB levels as well as NF-κB levels and activity and thereby performing anti-inflammatory and pro-fibrinolytic functions [144]. In cardiac fibroblasts stimulated with Angiotensin II to induce structural and functional remodeling mimicking that of heart failure, rivaroxaban diminished NF-κB activity by 82% [145].

Other non-antiplatelet and non-anticoagulant compounds also exhibit modulatory actions against NF-κB. For instance, a recent study demonstrated that nifedipine, a calcium channel blocker, triggers PPARβ/γ activity. Interestingly, PPARβ/γ inhibits NF-κB activation, hence attenuating intracellular Ca^2+^ mobilization, reducing inside-out αIIbβ3 signaling and fibrinogen binding, and preventing platelet aggregation. It is suggested that the inhibitory effect of PPARβ/γ on NF-κB activation following nifedipine treatment is mediated by its regulation of NO/cGMP/PKG1 [146]. Other drug activators of PPARs such as statins, fibrates [147], and thiazolidin [148,149] confer anti-aggregation properties seemingly by acting in a similar way and inhibiting NF-κB. A high dose of simvastatin for example significantly diminishes plasma low-density lipoprotein cholesterol (LDL) levels (including oxidized LDL) in parallel to the binding activity of NF-κB in PBMC [150].

Lastly, a few studies on natural molecules offer additional data on NF-κB modulation. Sesamol 3,4-methylenedioxyphenol), a constituent of sesame oil with antiplatelet effects, inhibits NF-κB pathway by inducing cAMP-PKA signaling cascade, which culminates in the inhibition of intracellular Ca^2+^ mobilization and ultimately platelet aggregation [151]. Vitamin C dose-dependently prevents IκB degradation and NF-κB activation in vitro, thus aiding to reduce inflammation [152]. Vitamin E, another antioxidant, shows similar effects in vitro and in vivo, although it is unclear whether the inhibition is direct or automatically follows a reduction in oxidative stress [153]. Other natural molecules such as β-carotene, *N*-acetylcysteine, selenium, and omega 3 fatty acids have also exhibited indirect inhibition of the NF-κB pathway [11].

Evident above, there is a scarcity of data on drug pharmacodynamics at the level of platelet NF-κB, as compared to NF-κB in other cells. However, this is fathomable, taking into consideration the reasoning of this review article, that is expanding our limited knowledge on the role of NF-κB in platelet functioning. Consequently, one can extrapolate from the studies performed thus far that targeting platelet NF-κB, as demonstrated with cellular NF-κB, might confer a plethora of therapeutic benefits first at the level of platelet survival, priming, activation, and aggregation and second in the context of atherothrombotic coronary artery diseases. Still and all, further research is mandatory to provide us with the facts.

## 6. The Interplay between mRNA, miRNA, and NF-κB in Platelets

As mentioned in Section 3, platelets harbor both mRNA and miRNA. In fact, RT-PCR and microarray studies of highly purified human platelets corroborate that the platelet proteome is a copy image of its transcriptome, having identified an average of 2500 platelet-expressed mRNA transcripts (approximately 12,500-fold less than nucleated cells) [39,154,155]. Likewise, Landry et al. have shown that human platelets encompass a wide array of miRNAs (miRs) [156], small (about 22 nucleotides) non-coding RNAs derived from long RNA precursors, and Gregory et al. [157] evidenced further a modulatory role of miRNA in controlling platelet activity [156]. Later genome wide profiling of platelet RNA allowed the identification of about 284 platelet-expressed miRNA [158,159]. Other studies provide further insight on the involvement of miRNA in platelet activation by showing a differential up- and down-regulation of 6 miRNAs upon thrombin stimulation [160].

The NF-κB pathway is not only a direct target but also a modulator of several miRNAs. For instance, miR-223, miR-199a, miR-155, and miR-124a inhibit the expression of different IKK subunits, thus preventing NF-κB activity. On the other hand, NF-κB also regulates a few miRNAs such as miR-29b which is implicated in tumor suppression [161]. It is well known that functional RNA, including miRNA, can move intracellularly via exosomal shuttle-like microparticulate vesicles released by the cell [162]. In addition, activated platelets possess the ability to release microvesicles harboring functional miRNA which can be passed to nucleated cells [163] to modulate their pathological states [164,165], which may suggest the presence of an orchestrated interplay between miRNA and NF-κB mediated by platelet microvesicles. This was recently manifested in a murine study on the protective mechanisms of action of thrombin-activated platelet-derived exosomes in atherosclerosis. In the scope of our topic, however, and considering that platelet NF-κB is found in platelet microparticles [164], one can postulate that in addition to its non-genomic role in platelet survival and activation-aggregation, platelet NF-κB may harbor an extra-platelet genomic role, which unfolds only following platelet microparticles uptake by other nucleated cells.

## 7. Conclusions

NF-κB has been shown to be implicated in the transcriptional regulation of over a hundred genes, a large number of these genes exhibit pro-inflammatory properties [166]. In nuclei-devoid platelets which play a major role in cardiovascular diseases, NF-κB is shown to have a different non-genomic function, as summarized in Figure 3. Herein, our endeavors in recapitulating diverse data on platelet NF-κB might help to characterize the exceptional function of this protein in platelets. Although emerging studies corroborate that NF-κB has a primordial role in positively regulating platelet survival, priming, activation, and aggregation, further investigations are warranted to fully elucidate the mechanics and roles of NF-κB in those acellular fragments, especially that another perplexing and equally interesting extra-platelet function for platelet NF-κB has recently been fading in and since few studies showcase an opposite role for NF-κB in platelet function and may, therefore, act as a double-edged sword [28,61]. This, and given the fact that the targeting of NF-κB by several active compounds is elucidated to ameliorate diverse pathophysiological conditions, more pre-clinical research might also bestow upon platelet NF-κB a therapeutic potential in cardiovascular diseases. Thus, inhibiting platelet NF-κB may have a high therapeutic potential to treat thrombotic disorders. Because platelet activation is linked to hemostasis, and also has a key role in inflammation and thrombosis, our present review demonstrates that inhibition of NF-κB interferes with platelet function by reducing its thrombogenic potential and holds great promise when compounds that block NF-κB activation are considered for treating various thrombo-inflammatory diseases.

## Figures and Tables

**Figure 1 ijms-20-04185-f001:**
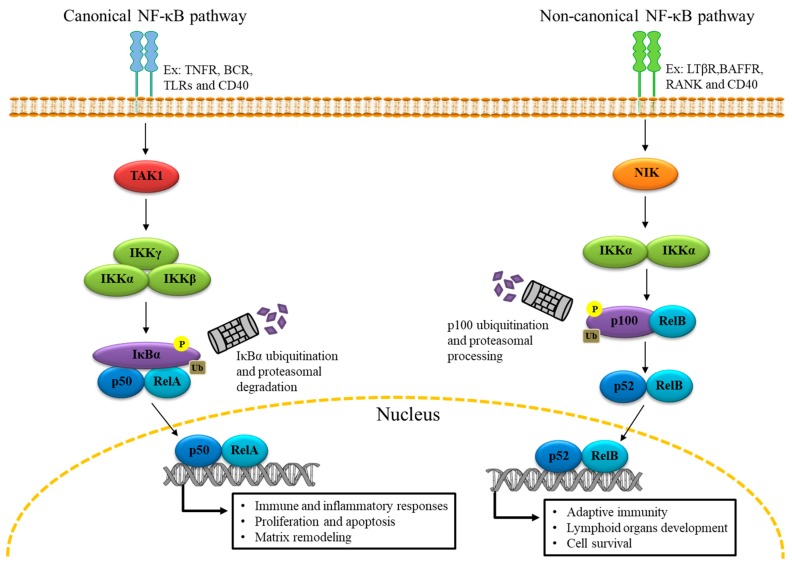
NF-κB Activation: Canonical and non-canonical pathways. The activation of the canonical pathway is triggered by various receptors such as the tumor necrosis factor receptor (TNFR) and Toll-like receptors (TLRs) or the B-cell receptor (BCR). This pathway involves activation of the IκB kinase (IKK) complex (IKKα, IKKβ, and IKKγ) by TAK1 and IKK-mediated IκBα phosphorylation. IκBα phosphorylation induces its ubiquitination and degradation by the proteasome leading to the nuclear translocation of p50/RelA dimers. The activation of the non-canonical pathway is activated by different receptors such as the lymphotoxin-β receptor (LTβR) and the B-cell activating factor receptor (BAFFR). This pathway relies on the activation of NF-κB-inducing kinase (NIK) an IKKα, which leads to the phosphorylation and ubiquitination of p100 and subsequently the processing of p100 by the proteasome to generate transcriptionally active p52/RelB dimers.

**Figure 2 ijms-20-04185-f002:**
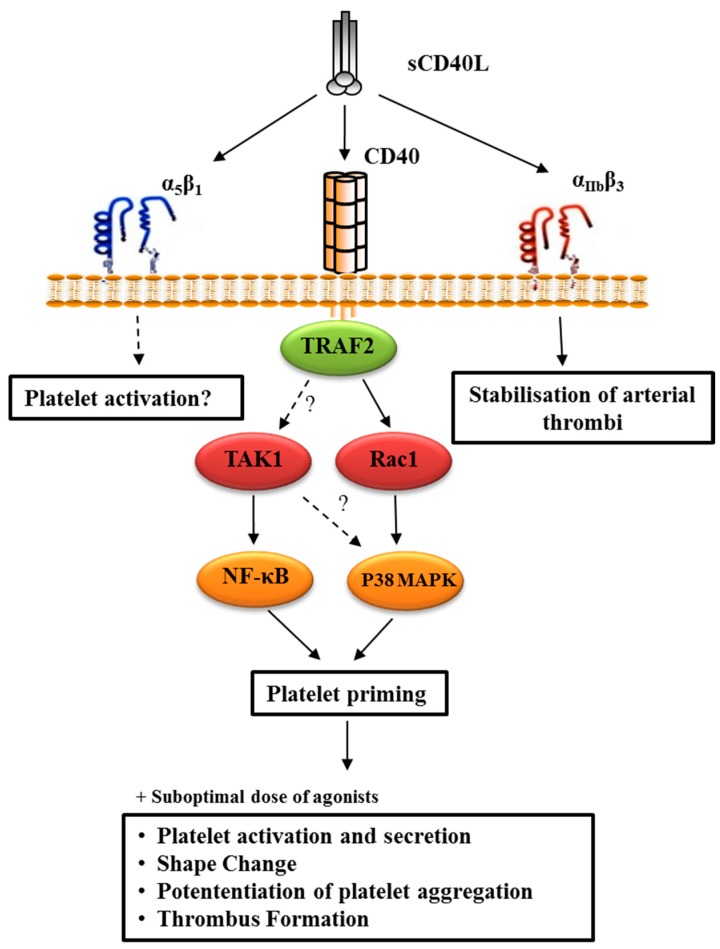
Proposed pathways of sCD40L/CD40 function in platelets. sCD40L enhancement of platelet reactivity or priming involves its main receptor CD40 binding to an adaptor protein the (TNFR)-associated factor-2 (TRAF2) and downstream signaling Rac1/p38-MAPK and TAK1/NF-κB. In response to suboptimal doses of agonists, primed platelets potentiate platelets function, which can promote thrombus formation.

**Figure 3 ijms-20-04185-f003:**
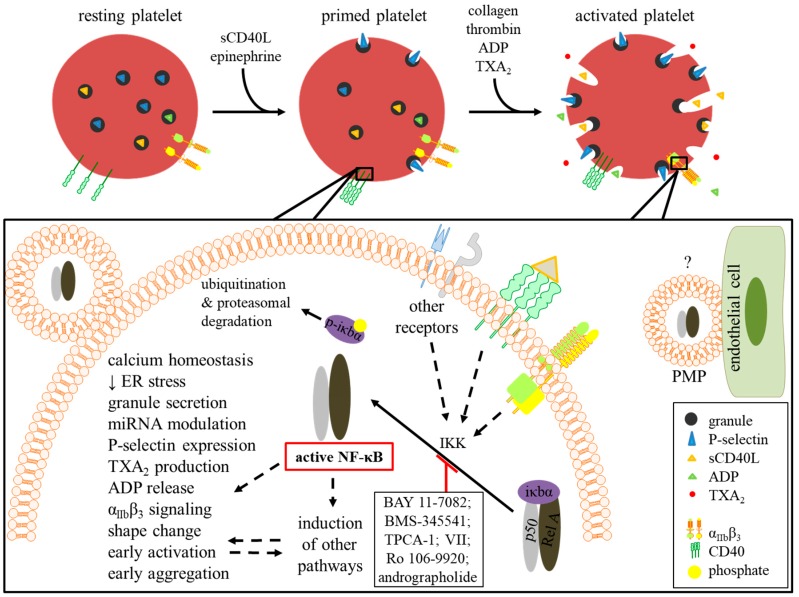
Schematic diagram portraying the role of NF-κB in platelets. Upon ligation of different receptors by priming (sCD40L and epinephrine) and/or activating ligands (collagen; thrombin; ADP; and TxA_2_), IKK activation triggers NF-κB pathway. Unlike nucleated cells-originating NF-κB, which translocates into the nucleus and binds genomic DNA, platelet NF-κB confers functions of other nature as shown by the utility of several pharmacological inhibitors of IKK such as BAY 11-7082 and BMS-345541. Activated NF-κB plays a role in platelet survival and platelet priming. Platelet NF-κB might also be involved in regulating miRNA, however, this requires further validation. Although it has been shown that platelet NF-κB carried by platelet microparticles (PMP) is endocytosed by other cells such as endothelial cells, its exact extra-platelet functions are still elusive.

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
