# Peer review of "Role of NF-κB in Platelet Function"

_ijms, 2019, doi:10.3390/ijms20174185_

Round 1

Reviewer 1 Report

In this manuscript, the authors comprehensively summarize the role of NF-kB in platelet function. Some points are listed below.

Section 4.1 platelet survival and apoptosis. Platelet life span is not only control by apoptosis pathway. It may also be regulated by GPIba, the shedding of GPIba or the GPIba-VWF association (PMID: 29475962; PMID: 27417583; PMID: 29097365). The authors discussed how platelet agonists (thrombin, collagen) result in the activation of NF-kB, but did not discuss how NF-kB activation influence platelet activity. Detailed description about how NF-kB affect platelet degranulation is useful.

Author Response

Response to Reviewer 1 Comments

Point 1: Section 4.1 platelet survival and apoptosis. Platelet life span is not only control by apoptosis pathway. It may also be regulated by GPIba, the shedding of GPIba or the GPIba-VWF association (PMID: 29475962; PMID: 27417583; PMID: 29097365).

Response 1: As requested by the reviewer, we added a section (lines144-154) discussing the regulation of platelet life span by the shedding of GPIba:

"Aside of the apoptosis pathway, inhibiting GPIbα shedding in platelets was able to reduce platelet clearance [55-57]. Indeed, the shedding of GPIbα is a physiological mechanism mediated by ADAM17 that take place constantly on platelet surface [58]. However, GPIbα shedding is linked to increase of platelet storage lesion. Indeed, under physiological shear stress, the binding of Von Willebrand Factor (VWF)  with GPIbα induces unfolding of the mechanosensory domain (MSD) on the platelet surface, thereby boosting shedding of GPIbα and triggering GPIb-IX signaling, leading to rapid platelet clearance [59,60]. NF-κB could contribute to GPIbα shedding in platelets; for instance, in response to thrombin stimulation, IKKβ-deficient platelets were unable to shed GPIbα. However, in response to ADP or collagen stimulation, GPIbα shedding was unaffected; this might suggest that IKKβ is uniquely implicated in thrombin-induced GPIbα shedding [61]. "

Point 2: The authors discussed how platelet agonists (thrombin, collagen) result in the activation of NF-kB, but did not discuss how NF-kB activation influence platelet activity. Detailed description about how NF-kB affect platelet degranulation is useful.

Response 2: In the original manuscript, we had described how NF-κB induce platelet degranulation. For more clarity, we added the highlighted paragraph to the discussion (lines 176-182)

Karim et al. further explain IKKβ implication in platelet secretion through its ability to phosphorylate synaptosome-associated protein-23 (SNAP-23), a member of membrane proteins complex called soluble N-ethylmaleimide-sensitive-factor attachment protein receptors (SNAREs) which regulates granule secretion. Indeed, SNAREs play a major role in the fusion of platelet granules with the platelet membrane. In fact, when platelets are activated in response to thrombin stimulation, vSNAREs (vesicular SNAREs) such as VAMP-8 (vesicle-associated membrane protein–8) that present on platelet granules surface bind to tSNAREs (target SNAREs) such as Syntaxin 11 expressed on platelet membrane. The binding is mediated mainly by SNAP-23. Thus, SNAP-23 phosphorylation in thrombin-stimulated platelets facilitate the fusion between granules and plasma membrane for cargo release [66]. Opposing, the use of selective inhibitors of IKKβ (BMS-345541[67], TPCA-1 [68], and BAY-11-7082) prevents thrombin-stimulated SNAP-23 phosphorylation in a dose-dependent manner [66]. Additionally, the same group shows that thrombin-stimulated platelets from IKKβ knockouts have a lower capacity to release alpha, dense, and lysosomal granules content by an overall of 30%. Wei et al. [61] who demonstrated that platelet IKKβ deficiency exhibits decreased secretion and activation confirmed this observation. Interestingly, a recent study shows that the inhibition of MALT1, an upstream regulator of IKK complex and a member of the functional proteasome CARMA/MALT1/Bcl10 (CBM) complex, prevents platelet activation and secretion by abrogating SNARE formation [69]. Such findings suggest that NF-κB could contribute to platelet degranulation in platelet secretion.

Reviewer 2 Report

The review is very readable and has summarized representative and significant publications in the field. Here are some suggestions for the reference.

1. The author discussed NF-kB in the beginning; in the later part, the discussion of the activation of platelet is included. I feel it will might be good to include a cartoon to demonstrate Nf-kB activation pathway including the classical activation and the alternative cascade in section 2. After all, Nf-kB is a key ingredient here and not everyone is familiar with it.

2. I understand the key point of this manuscript is to provide information of NF-kB signaling in platelet, I feel since the author mentioned the fact that the platelet plays a major role in cardiovascular diseases in several places, it might be informative for readers, who are not familiar with the platelet field, to know why platelet is important to cardiovascular diseases in general, what other factors/signaling are important to regulate its function, and finally why we care NF-kB signaling, but not others. There are many details of NF-kB and platelet included, but I feel I might lose the big picture.

3. In the conclusion section, it might be informative and will help to remind readers where the gaps/controversial results are and which future directions should be aimed. I feel a little bit lost in the end why I should care about NF-kB in platelet, but not other pathways. Is that because it is more important than other pathways or if there is any controversial results.

4. The author might also consider to cite the paper below:

Lu WJ, Lin KH, Hsu MJ, Chou DS, Hsiao G, Sheu JR. Suppression of NF-κB signaling by andrographolide with a novel mechanism in human platelets: regulatory roles of the p38 MAPK-hydroxyl radical-ERK2 cascade. Biochem Pharmacol. 2012; 84:914–924.

Author Response

Response to Reviewer 2 Comments

Point 1: The author discussed NF-kB in the beginning; in the later part, the discussion of the activation of platelet is included. I feel it will might be good to include a cartoon to demonstrate Nf-kB activation pathway including the classical activation and the alternative cascade in section 2. After all, Nf-kB is a key ingredient here and not everyone is familiar with it.

Response 1: As requested by the reviewer, a cartoon of the canonical and non-canonical pathways of activation of NF-κB was added as Figure 1, lines 89 - 97

Point 2:  I understand the key point of this manuscript is to provide information of NF-kB signaling in platelet, I feel since the author mentioned the fact that the platelet plays a major role in cardiovascular diseases in several places, it might be informative for readers, who are not familiar with the platelet field, to know why platelet is important to cardiovascular diseases in general, what other factors/signaling are important to regulate its function, and finally why we care NF-kB signaling, but not others. There are many details of NF-kB and platelet included, but I feel I might lose the big picture. 

Response 2: The following highlighted paragraphs were added in the introduction to clarify platelet involvement in cardiovascular diseases as well as to highlight the importance of investigating NF-κB signaling in thrombo-inflammatory diseases.

Lines 28-49:

"Cardiovascular disease is the single largest cause of death globally. According to the report of the World Health Organization (WHO), the death rate caused by cardiovascular diseases worldwide is estimated at 17.3 million people in 2008, accounting for 30% of deaths globally. In addition, it is estimated that this number may increase to 23.6 million people [1]. Cardiovascular disease is often caused by thrombotic events such as coronary heart disease. Platelets are major players in the occurrence of cardiovascular diseases since they are involved in various thrombo-inflammatory diseases, particularly atherosclerosis and its progression to atherothrombosis in acute coronary syndrome (ACS) patients [2-5]. Indeed, platelets mediate primary hemostasis and formation of thrombosis, however, thrombosis can become pathological when it occurs, for example, after the rupture of an atherosclerotic plaque. Thus, following an atherosclerotic vascular lesion, exposed sub-endothelial collagen, alongside thrombin and adenine diphosphate (ADP) production triggers platelet activation by interacting with several receptors expressed on platelets such GPIb-IX-V, GPVI-FcRγ, protease activated receptors (PARs), P2Ys and integrins (α2β1, α5β1, α6β1, αvβ3 et αIIbβ3).  Consequently, the "inside-out" signaling of the αIIbβ3 integrin, in response to platelet activation, leads to platelet aggregation thanks to the binding of αIIbβ3 to fibrinogen promoting the formation of a thrombus [6,7]. Such process could induce a partial or complete occlusion of the blood vessel, which leads to a decrease or blockage of the blood flow and thus becomes a cause of occurrence of ischemia or infarction of an irrigated organ such as the heart. Furthermore, the release a plethora of inflammatory mediators by activated platelets such as soluble P-selectin, soluble CD40 ligand (sCD40L, also known as CD154) and interleukin-1 beta (IL-1β), which interact with cells that mediate inflammation such as circulating leukocytes, endothelial cells, and progenitor cells, thereby aggravating inflammatory responses [8,9]."

Lines 55-65:

" Indeed, the activation of NF-κB in endothelial cells in response to an inflammatory environment triggers the expression of adhesion molecules, which increases binding and transmigration of leukocytes and platelets, while unleashing their thrombogenic potential. The activation of NF-κB in monocytes contributes to the expression of tissue factor and release of inflammatory cytokines and is required for their differentiation to macrophages. In neutrophils, NF-κB activation extend their survival and induce the expulsion of neutrophil extracellular traps (NETs), which exert antibacterial functions and triggers a strong coagulatory response and may induce formation of microthrombi. NF-κB plays a crucial role in in lymphocyte proliferation and cytokine and chemokines production, which promote enhanced inflammatory and thrombotic response by recruiting platelets. In this review, since NF-κB is a central hub in thrombo-inflammatory reactions."  

Point 3: In the conclusion section, it might be informative and will help to remind readers where the gaps/controversial results are and which future directions should be aimed. I feel a little bit lost in the end why I should care about NF-kB in platelet, but not other pathways. Is that because it is more important than other pathways or if there is any controversial results.

Response 3: As requested, the conclusion was modified to clarify this issue, by adding the following:

Lines 481-482:

"NF-κB has been shown to be implicated in the transcriptional regulation of over a hundred genes, a large number of these genes exhibit pro-inflammatory properties [166]. "

Lines 490-491:

" since few studies showcase an opposite role for NF-κB in platelet function and may therefore act as a double-edged sword [28,61]. "

Lines 494-498:

" Thus, inhibiting platelet NF-κB may have a high therapeutic potential to treat thrombotic disorders. Because platelet activation is linked to hemostasis, and also has a key role in inflammation and thrombosis, our present review demonstrates that inhibition of NF-κB interferes with platelet function by reducing its thrombogenic potential and hold great promise when compounds that blocks NF-κB activation are considered for treating various thrombo-inflammatory diseases. "

Point 4: The author might also consider to cite the paper below:

Lu WJ, Lin KH, Hsu MJ, Chou DS, Hsiao G, Sheu JR. Suppression of NF-κB signaling by andrographolide with a novel mechanism in human platelets: regulatory roles of the p38 MAPK-hydroxyl radical-ERK2 cascade. Biochem Pharmacol. 2012; 84:914–924.

Response 4: This paper was already cited in our original manuscript  as reference # 45 at line 113. This reference was also cited in the revised manuscript at line 339.

Round 2

Reviewer 1 Report

The authors addressed my previous concerns. I have no further questions.

Author Response

Comments to reviewer 1

We are glad our comments satisfied your previous concerns and we thank you for improving our manuscript.